# A Plasma Pyrophosphate Cutoff Value for Diagnosing Pseudoxanthoma Elasticum

**DOI:** 10.3390/ijms25126502

**Published:** 2024-06-13

**Authors:** Isabelle Rubera, Laetitia Clotaire, Audrey Laurain, Alexandre Destere, Ludovic Martin, Christophe Duranton, Georges Leftheriotis

**Affiliations:** 1University Côte d’Azur, CNRS, LP2M, Labex ICST, 06107 Nice, France; isabelle.rubera@univ-cotedazur.fr (I.R.); laetitia.clotaire@etu.univ-cotedazur.fr (L.C.); audrey.laurain@aphp.fr (A.L.); leftheriotis.g@chu-nice.fr (G.L.); 2Pharmacology Department, Nice University Hospital, 06000 Nice, France; 3FAVA-MULTI South Competence Center for Rare Arterial Calcifying Disease, Nice University Hospital, 06000 Nice, France; 4PXE Reference Center, MAGEC Nord, Angers University Hospital, 49000 Angers, France; lumartin@chu-angers.fr

**Keywords:** diagnosis, rare disease, PXE, threshold, PPi, PPi quantification

## Abstract

Pseudoxanthoma elasticum (PXE) is a rare inherited systemic disease responsible for a juvenile peripheral arterial calcification disease. The clinical diagnosis of PXE is only based on a complex multi-organ phenotypic score and/or genetical analysis. Reduced plasma inorganic pyrophosphate concentration [PPi]p has been linked to PXE. In this study, we used a novel and accurate method to measure [PPi]p in one of the largest cohorts of PXE patients, and we reported the valuable contribution of a cutoff value to PXE diagnosis. Plasma samples and clinical records from two French reference centers for PXE (PXE Consultation Center, Angers, and FAVA-MULTI South Competent Center, Nice) were assessed. Plasma PPi were measured in 153 PXE and 46 non-PXE patients. The PPi concentrations in the plasma samples were determined by a new method combining enzymatic and ion chromatography approaches. The best match between the sensitivity and specificity (Youden index) for diagnosing PXE was determined by ROC analysis. [PPi]p were lower in PXE patients (0.92 ± 0.30 µmol/L) than in non-PXE patients (1.61 ± 0.33 µmol/L, *p* < 0.0001), corresponding to a mean reduction of 43 ± 19% (SD). The PPi cutoff value for diagnosing PXE in all patients was 1.2 µmol/L, with a sensitivity of 83.3% and a specificity of 91.1% (AUC = 0.93), without sex differences. In patients aged <50 years (i.e., the age period for PXE diagnosis), the cutoff PPi was 1.2 µmol/L (sensitivity, specificity, and AUC of 93%, 96%, and 0.97, respectively). The [PPi]p shows high accuracy for diagnosing PXE; thus, quantifying plasma PPi represents the first blood assay for diagnosing PXE.

## 1. Introduction

Pseudoxanthoma elasticum (PXE) is a chronic and disabling monogenic rare disease caused primarily by mutations in the *ABCC6* gene [1]. Its prevalence has been estimated between 1/25,000 and 1/100,000 in the general population, with female predominance [2]. PXE is characterized by abnormal ectopic calcifications in soft connective tissues such as the skin, retina and cardiovascular system. The clinical diagnosis of PXE [3,4] is currently based on clinical criteria (PHENODEX score [5]), which include the presence of 1 or 2 major ocular lesions (i.e., peau d’orange and/or angioid streaks) associated with 1 or 2 skin and/or mucosal lesions (i.e., pseudo-xanthomatous papules of the flexion folds) and/or a positive skin biopsy taken from the lesional skin, the neck or the flexural area in the absence of suggestive skin changes. Genetic diagnosis is a powerful method to detect PXE [6] but may be unsuccessful since a small but significant level of PXE patients have no *ABCC6* mutations. PXE demonstrates considerable inter- and intrafamilial heterogeneity and, in some cases, overlaps with other calcifying diseases, such as generalized arterial calcification in infancy (GACI) [7], that is linked to heterozygous pathogenic mutations of the *ENPP1* gene [8]. Moreover, a PXE-like phenotype was also described due to pathogenic variants in the *GGCX* gene that encoded the gamma-glutamyl-carboxylase protein with alteration of anti-calcification processes [9]. Therefore, genetic diagnosis is presently reserved for atypical cases.

Due to the high phenotypic and clinical variability of PXE, there is a growing need to improve diagnostic accuracy due to prognostic implications and future development of therapeutic options for this disease.

The *ABCC6* gene involved in PXE encodes the ATP-binding cassette transporter ABCC6, the activity of which appears to be linked to the cellular outflow of ATP and the subsequent generation of inorganic pyrophosphate (PPi) [10]. Pyrophosphate is known as a physiological plasma anti-calcifying factor, and plasma PPi is chronically decreased in *Abcc6*^−/−^ animal models [10,11,12] and by 25 to 60% in PXE patients [12,13,14,15,16,17]. To date, in the absence of standardized measurement methods to determine PPi concentration in the plasma, the diagnostic value of PPi for PXE has never been tested, and its relevance remains to be established.

Herein, we develop a new method combining enzymatic and ion chromatographic techniques to determine the plasma PPi concentration with accuracy, allowing to report a diagnostic value in one of the largest European cohorts of PXE patients.

## 2. Results

### 2.1. A Significant Decrease of Plasma PPi Concentration in a PXE Cohort

For this study, we have used a new method coupling two different technical approaches (enzymatic and ion chromatography), allowing a cross-control to accurately determine the PPi concentration in plasma samples.

In the non-PXE control cohort (46 healthy volunteers, (mean ± SD) age 45 ± 15 years; 55% women), the mean plasma PPi concentration was 1.61 ± 0.33 µmol/L, while in the PXE cohort (153 patients, 49 ± 15 years; 107 (71%) women), it was 0.93 ± 0.30 µmol/L (Figure 1). The PXE patients had a 43 ± 19% reduction in PPi concentration compared to non-PXE patients.

### 2.2. Determination of a PPi Cutoff Value for PXE Diagnosis

The statistical analysis plan was used, and the main results of this study are presented in Figure 2. Briefly, two ROC analyses were performed with the training dataset, comprising all PXE and with a training dataset of <50 years old PXE patients (<50 y.o., Figure 3).

A PPi cutoff (Youden index) for diagnosing PXE in all patients was calculated at 1.2 µmol/L, with a sensitivity of 83.3%, a specificity of 91.1%, and an AUC of 0.93 (Figure 2 and Figure 3A, Table 1).

**Figure 2 ijms-25-06502-f002:**
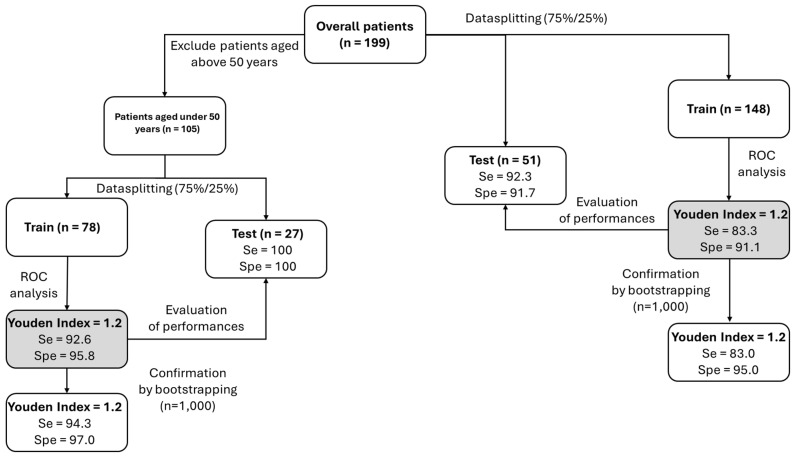
Proposed model for accurate determination of the PPi cutoff for PXE diagnosis. The overall population (n = 199) was split into training (75%, n = 148) and test (25%, n = 51) datasets. Cutoff values were determined by ROC analysis of the “training” dataset and of the “simulated-training” dataset (n = 1000, using a bootstrap analysis of the “training” dataset). The same strategy was used to determine the cutoff value in patients aged <50 y.o. Se: sensitivity; Spe: specificity.

**Figure 3 ijms-25-06502-f003:**
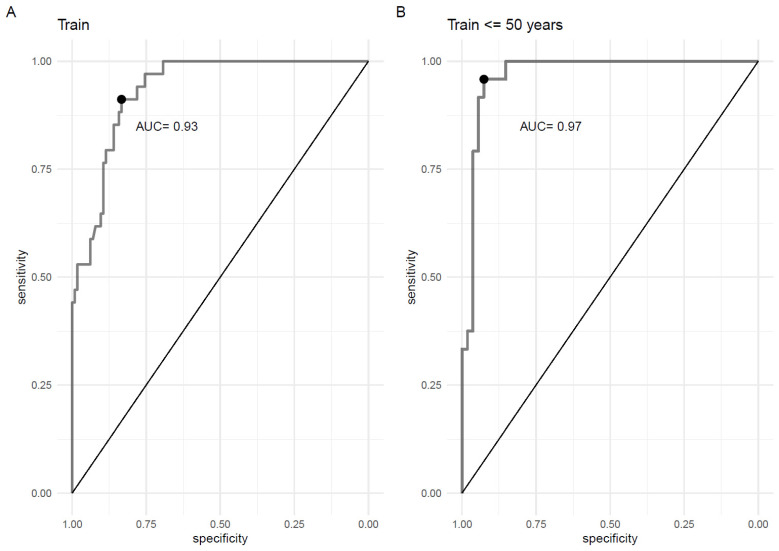
Determination of the best cutoff value of plasma inorganic pyrophosphate (PPi) for diagnosing PXE using receiver operating characteristic (ROC) curves in all patients (**A**) and in patients < 50 years old. (**B**). The black dots indicate the best compromise between sensitivity and specificity (Youden index). AUC: area under the curve.

**Table 1 ijms-25-06502-t001:** Performance of the determined thresholds (100% sensitivity, Youden index, and 100% specificity) in the “training” and “test” datasets for all patients and patients aged <50 y.o.

Threshold	Overall Patients				Patients Aged under 50 Years (<50 y.o.)			
	Train (n = 148)		Test (n = 51)		Train (n = 78)		Test (n = 27)	
	Sensitivity (%)	Specificity (%)	Sensitivity (%)	Specificity (%)	Sensitivity (%)	Specificity (%)	Sensitivity (%)	Specificity (%)
<1.70	100.0	44.1	100.0	33.3	100.0	33.3	100.0	37.5
**<1.20**	**83.3**	**91.1**	**92.3**	**91.7**	**92.6**	**95.8**	**100.0**	**100.0**
<1.10	69.3	100.0	79.5	100.0	85.2	100.0	89.5	100.0

With respect to the “simulated-training” dataset, the cutoff was 1.2 µmol/L, leading to a sensitivity of 83.0% and a specificity of 95.0% (Figure 2).

Analysis of patients aged <50 years old, the approximate age at which PXE should be diagnosed before the onset of complications, revealed that the PPi cutoff value was 1.2 µmol/L, with a sensitivity, specificity, and AUC of 92.6%, 95.8%, and 0.97, respectively (Figure 3B and Table 1). With respect to the “simulated-training” dataset, the cutoff was 1.2 µmol/L, with a sensitivity of 94.3% and a specificity of 97% (Figure 2).

The performance of the “test” dataset for all patients and patients aged <50 y.o. was (sensitivity/specificity) 92.3%/91.7% and 100%/100%, respectively (Figure 2).

The performances of the different thresholds (100% sensitivity threshold, Youden index, and 100% specificity threshold) in the overall population and patients aged <50 y.o. in the training and test dataset are presented in Table 1. The PPi cutoffs yielding 100% specificity (1.1 µM) or 100% sensitivity (1.7 µM) are identical in the overall population and in those aged <50 y.o. (Table 1).

### 2.3. Influence of Sex on the PPi Cutoff for PXE Diagnosis

The PPi cutoff value in the male and female “training” dataset aged <50 years old was 1.2 µmol/L for both men (sensitivity 94.4%, specificity 100%) and women (92.9% and 93.9%, respectively).

## 3. Discussion

Our study reports a cutoff plasma PPi value for diagnosing PXE, an inherited disease characterized by chronic disability due to extensive and progressive ectopic calcifications in connective tissues. Since 1994, the diagnosis of PXE has been based only on clinical symptoms, skin biopsy results, and family history [8]. Due to considerable inter- and intrafamilial phenotype heterogeneity, the involvement of different organ systems in this disease may greatly differ. The reasons for this phenotypic heterogeneity currently remain unknown, and attempts to establish genotype/phenotype correlations have failed to establish a consistent one [18] except in one study [19].

A low plasma PPi has been consistently reported in PXE patients with a reduction rate ranging from 25 to 60% [12,13,14,15,16,17] despite significant differences in the absolute values: plasma PPi levels in PXE patients versus healthy controls were reported in a Belgian cohort (mean value 0.497 vs. 0.985 µM [17]), in a French cohort (median value 0.77 vs. 1.53 µM [15]), in a Dutch cohort (0.53 vs. 1.13 µM [14]) and in a Spanish cohort (1.11 vs. 1.43 μM [16]). Altogether, comparisons of plasma PPi concentration between all these studies are somewhat difficult since it depends on (1) the type of blood collection tube (CTAD, citrate, heparin, EDTA, …), (2) the pre-analytic preparation of the plasma samples (i.e., ultrafiltration, platelet-depletion), and (3) the method used to measure PPi. A recent study [17] pointed out that PPi level does not correlate with the genotype or phenotype of PXE disease. However, we have already reported a strong correlation between age, arterial calcification, and disease severity [15], suggesting that time of exposure to a low plasma PPi concentration is a major determinant of arterial calcification and clinical severity in PXE.

According to our study, a plasma PPi cutoff value of 1.2 µmol/L detected almost 83% of PXE patients with high specificity (91%) and was not influenced by sex. Interestingly, when the population age was limited to <50 y.o. (i.e., the age period when PXE is usually diagnosed), the plasma PPi cutoff value was still 1.2 µmol/L, while the sensitivity and specificity increased to 93 and 96%, respectively. Therefore, PPi represents a promising biological blood marker that provides additional diagnostic criteria and reliable value for diagnosing PXE. To date, the use of plasma PPi for PXE diagnosis has never been established due to the lack of “gold standard” methods and technical limitations in precisely determining PPi concentrations, which are both overcome using our method.

Although the number of PXE patients included in this study is small, it remains among the largest since PXE is a rare disease. The threshold validation was performed on a small subset of the internal validation cohort, which can limit the performance of the external validation cohort. In addition, the use of the cutoff value is limited to our method, which combines enzymatic and ion chromatography techniques as a cross-control, which is not the case for the other tests usually carried out in laboratories. Therefore, access to this assay method in clinical practice remains to be developed.

In conclusion, our results show that the plasma PPi concentration may reliably contribute to PXE diagnosis. Furthermore, this blood test will significantly contribute to the development and validation of future treatments for this disease by clinical trials. A strategy currently being explored by the medical community is based on normalizing plasma PPi concentration in PXE patients [20]; therefore, it is essential to monitor PPi changes accurately over time during treatment.

## 4. Materials and Methods

### 4.1. Study Population

Medical records and biological samples collected between 2018 and 2023 from PXE patients were obtained from the biobank of the National PXE Reference Center (MAGEC Nord, Angers University Hospital, Angers, France) as part of the protocol for phenotyping the French PXE cohort (ClinicalTrials.gov Identifier: NCT01446380) and by the FAVA-MULTI South Competence Center for Rare Arterial Calcifying Disease, Nice University Hospital, Nice, France (ClinicalTrials.gov Identifier: NCT04868578). PXE diagnosis was performed by experienced clinicians from the two centers (LM and GL) on the basis of the actual diagnostic criteria for PXE [8,10]. Non-PXE patients (healthy volunteers) were recruited from routine clinical check-ups as controls. All PXE and non-PXE patients provided informed consent for the study.

### 4.2. Preanalytical Preparation of Blood Samples

Blood samples were collected in CTAD tubes on the morning after overnight fasting. These tubes were kept on ice until centrifugation (1000× *g*, 15 min, 4 °C). Plasma samples were collected and ultrafiltered at 4 °C using Amicon 0.5 mL Ultracel^®^ filters (Sigma-Aldrich, Saint-Quentin Fallavier, France) to eliminate platelets and 95% of the proteins (including albumin). All the preanalytical steps were performed as quickly as possible to avoid PPi degradation or blood cell lysis.

### 4.3. Plasma PPi Concentration Measurement

The plasma PPi concentration was determined by a new method combining enzymatic and ion chromatography (IC) techniques. This method allows cross-control of the measured PPi values and increases the reliability of the result.

Enzymatic method: Quantification of the PPi concentration in ultrafiltered plasma was performed using a modified method based on a previously described enzyme assay [5]. PPi was converted to ATP using sulfurylase (Bio-Techne, Rennes, France) in the presence of APS (adenosine-5′-phosphosulfate; Sigma-Aldrich, Saint-Quentin Fallavier, France) after incubating for 40 min at 37 °C and subsequently for 10 min at 90 °C to inactivate sulfurylase. The generated ATP was quantified using an ATP Determination Kit (ATPlite; PerkinElmer, Courtabeuf, France). The luminescence was measured on a microplate reader (Synergy HT, BioTek, Shoreline, WA, USA). PPi values were obtained by subtracting the basal ATP level measured with ATP sulfurylase-inactivated solution in each sample.

Ion chromatography method: Patient plasma ultrafiltrates were deproteinized using acetonitrile (1/1). The samples were strongly mixed and centrifuged (12,000× *g*, 10 min, 4 °C). The same protocol (acetonitrile) was also used to quantify the calibration standard solutions (sodium pyrophosphate dibasic, Santa Cruz Technologies, Heidelberg, Germany). PPi detection by conductimetry was performed using an ion chromatographic Dionex ICS-5000 plus system (Thermo Scientific, Courtabeuf, France). The system was equipped with an eluent generator (EGC500KOH), a guard precolumn (AG11-HC), and an analytical column (IonPac AS-11-HC). PPi peak quantification was performed using “Chromeleon software version 7.2 SR4” (Thermo Scientific, Courtabeuf, France) by measuring the surface area, and the results were compared to a PPi standard curve.

### 4.4. Statistical Analysis

All the statistical analyses were performed in Prism 10.0 (GraphPad) or R software version 4.3.2 [21]. Data are presented as mean ± SD (standard deviation). Statistical differences between PXE patients and non-PXE patients were compared using the Mann–Whitney test.

The cutoff values for the plasma PPi concentration were determined using the receiver operating characteristic (ROC) curve method to determine the best compromise between sensitivity and specificity (Youden index) and also to explore the performance of other thresholds (100% sensitivity and 100% specificity). The ROC analysis was performed using the R package version 1.18.5 “pROC” [22].

The diagnostic performance of a given PPi cutoff value was tested after the overall population was split into “training” (75%, n = 148) and “test” (25%, n = 51) datasets. The cutoffs determined by the ROC analysis were subsequently compared to a cutoff determined using a bootstrap analysis issued from the “training” set (i.e., “simulated-training” dataset = 1000). Finally, the cutoff was tested in the “test” dataset. A similar analysis was performed to determine the cutoff value for patients <50 years old (<50 y.o.). The effect of sex on the cutoff was also determined in the “simulated-training” set and further applied to the test dataset to evaluate the diagnostic performance as a function of sex.

## Figures and Tables

**Figure 1 ijms-25-06502-f001:**
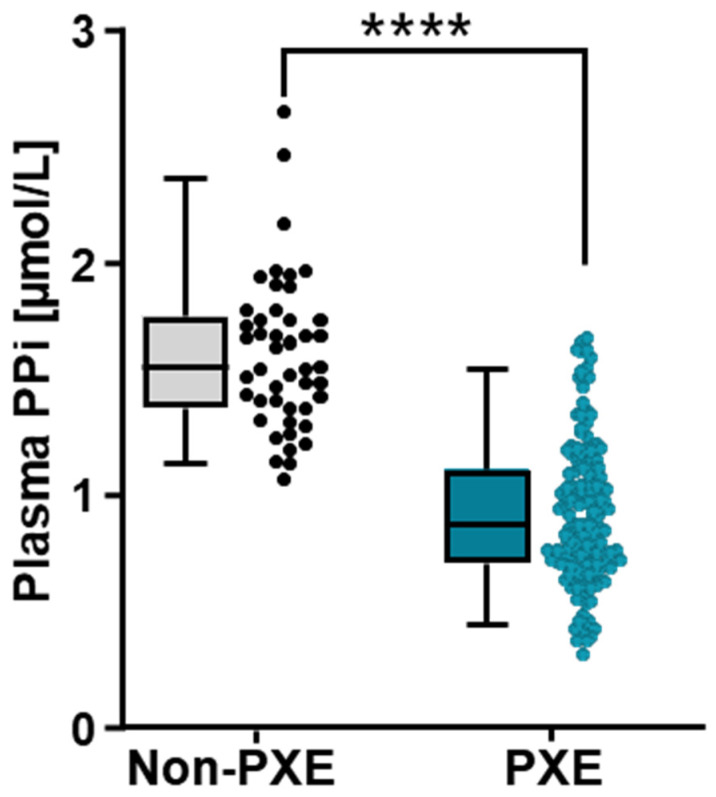
Plasma PPi concentrations measured in non-PXE (n = 46) and PXE patients (n = 153). Solid bars are the median values, and whiskers show the interquartiles (5/95). **** *p* < 0.0001.

## Data Availability

The datasets generated during and/or analyzed during the current study are available from the corresponding author upon reasonable request. Biological samples for the study are stored and available in the Angers’ biobank under the reference BB-0033-00038.

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
