# Peer review of "A Plasma Pyrophosphate Cutoff Value for Diagnosing Pseudoxanthoma Elasticum"

_ijms, 2024, doi:10.3390/ijms25126502_

Round 1

Reviewer 1 Report

Comments and Suggestions for Authors

1. The main problem is that the authors must justify the advantages of their study concerning previous studies and the new contributions of this manuscript concerning other methods. Furthermore, the authors need to describe their results and make better use of figures and tables since the way the manuscript is presented is descriptive.

2. Should compare the results obtained with previous results from the literature. Additionally, it must be included in the main document if the results obtained with your method correlate with the PXE diagnosis performed by experienced clinicians from the two centers (LM and GL).

3. Figures need to be described in the main document. Authors should use their Figures better since they are only defined once in the entire document and do not give a detailed explanation of it. The table must also be described. They are only mentioned, and there is no detailed description of them in the main document. Please include this information.

4. In the introduction, the authors should provide more information regarding studies in patients in which a decrease in PPi concentration was detected.

5. A paragraph is requested to be introduced where reference is made to the disease and its prevalence. Must provide this information in the introduction section

Specific comments

1. Abstract section: 1. The authors mention, "Herein, we report the diagnostic value of the plasma PPi concentration in a large European cohort of PXE patients. This is incorrect; in fact, one of the problems of this study is the sample size.

Results

2. In the results section, a short introduction and discussion in each section must be provided before describing all the experiments and results that support each section. This is to help the reader understand the importance of the results.

3. Line 66. Are the non-PXE control cohort (46 volunteers) healthy? Please indicate.

4. I suggest making a more detailed description in the text since the figure contains elements that are not mentioned. 

5. Add to the legend figure how SD was determined.

6. It is necessary to indicate that other methods exist to validate a diagnostic method and determine a cutoff value. Is there only ROC analysis? More information on the cutting analyses will be provided.

7. Figure 2. What does se, spe, mean? Please define it in figure legends.

8. Please describe in the main document the 2.3 Influence of sex on the PPi cutoff for PXE diagnosis.

9. The discussion section needs to be improved.

10. Limitations: in addition to the limitations, it must be included that the number of patients is small.

11. The conclusion should be improved. The authors mention “significantly contribute to the development and validation of future treatments for this disease.” Please explain how these results contribute to future treatments for this disease.

12. The authors should mention in the introduction and results that they used combinations of enzymatic and ion chromatography techniques as a cross-control.

Author Response

First, we would like to thank the two reviewers who took time to assess our manuscript and for the critical and constructive feedback they provided. Following those comments, we revised our manuscript to address their concerns and we hope that this study is now suitable for publication in IJMS. Changes are marked up in red in the revised version and a point-by-point response to the reviewers is provided below in blue.

Reviewer 1

  1. The main problem is that the authors must justify the advantages of their study concerning previous studies and the new contributions of this manuscript concerning other methods. Furthermore, the authors need to describe their results and make better use of figures and tables since the way the manuscript is presented is descriptive.

Our study is the first to our knowledge to establish a threshold value for the diagnosis of PXE. At present, there is no biological criterion for the diagnosis of PXE. Indeed, previous studies have showed that plasma PPi concentration is chronically decreased in PXE by ~25 to 60% (Sanchez-Tevar et al. Annals of translational Med. 2019, Jansen et al. PNAS 2014), but a cut-off value has never been reported for the pathological level of PPi in these patients. Determining a cutoff could be a very contributive mean to help the clinicians especially when the clinical phenotype is not fully relevant. Of course, our study is mostly descriptive since the mechanistic origin of the low PPi level remains elusive for the moment.

We thank the reviewer and have done our best to improve the description of our results in the revised version.

  1. Should compare the results obtained with previous results from the literature. Additionally, it must be included in the main document if the results obtained with your method correlate with the PXE diagnosis performed by experienced clinicians from the two centers (LM and GL).

We thank the reviewer for his/her remark. Unfortunately, we cannot compare the results with previous results since none of these studies were performed with the same experimental conditions. There is a big variability in the PPi dosage that blunt the comparisons between published data. A good example of this discrepancy is illustrated in a recent paper from Hsu et al (10.1093/rheumatology/keab508) were the baseline PPi values from 2 cohorts from 2 different centers (US and Europe) could not be compared due to laboratory differences. For these reasons, the percentage of changes from controls versus patients can only be compared but cannot be a threshold since the linearity of each dosage method cannot be determined. This is to overcome this problem that we decided to provide a threshold value using our method. We have now discussed this point in the discussion section. The patients were diagnosed by experienced clinicians from the two centers (LM and GL) and then we proceed to PPi quantification. 26 PXE over 153 exhibited PPi concentration higher than the cutoff we determined as illustrated in the Fig 1.

  1. Figures need to be described in the main document. Authors should use their Figures better since they are only defined once in the entire document and do not give a detailed explanation of it. The table must also be described. They are only mentioned, and there is no detailed description of them in the main document. Please include this information.

We have now described a bit more the different figures and slightly extended the results section, but the referee should keep in mind that the format is limited for a “communication” in IJMS.

  1. In the introduction, the authors should provide more information regarding studies in patients in which a decrease in PPi concentration was detected.

As requested, we have added more information regarding studies in which a decrease in PPi concentration was detected in PXE patients.

  1. A paragraph is requested to be introduced where reference is made to the disease and its prevalence. Must provide this information in the introduction section

We have now introduced epidemiology data about PXE in the introduction section.

Specific comments

  1. Abstract section: 1. The authors mention, "Herein, we report the diagnostic value of the plasma PPi concentration in a large European cohort of PXE patients. This is incorrect; in fact, one of the problems of this study is the sample size.

Our cohort includes more than 150 PXE patients and PPi plasma concentration was measured in all of them. To our knowledge this is one the most important cohort of PXE patients with individual PPi values. Referee 1 must be aware that PXE is a rare disease with a low prevalence. In the abstract we pointed out that the French PXE is “one of the largest cohorts of PXE” and the number of 153 used in this study was also given without qualifying the cohort as “large”. This was the case in the introduction but we now removed this qualifying adjective.

Results

  1. In the results section, a short introduction and discussion in each section must be provided before describing all the experiments and results that support each section. This is to help the reader understand the importance of the results.

We paid attention to this comment and the text has been modified accordingly.

  1. Line 66. Are the non-PXE control cohort (46 volunteers) healthy? Please indicate.

Non-PXE control patients are healthy subjects. This information has been added in the result section and throughout the text.

  1. I suggest making a more detailed description in the text since the figure contains elements that are not mentioned. 

We paid attention to this comment and the text has been modified accordingly.

  1. Add to the legend figure how SD was determined.

Data are presented in the text as mean ± SD (standard deviation). This information was added in the Statistical Analysis paragraph of the Materials and Methods section.

  1. It is necessary to indicate that other methods exist to validate a diagnostic method and determine a cutoff value. Is there only ROC analysis? More information on the cutting analyses will be provided.

We agree with the referee 1 about the existence of alternative of ROC analysis for instance Precision-Recall curve or confusion matrix to evaluate the PPV or NPV (positive or negative value). We also used confusion matrix to estimate the SE and SPE in the test dataset, but we have chosen not to present the results. The ROC analysis was carried out to determine the best compromise between SE and SPE (Youden index) but also to explore the performance of other thresholds. For example, a threshold of 100%SE or 100%SPE. The PR curve was used to assess the impact of our threshold on the number of false positives, but this is of little importance as genetic analyses are carried out in case of doubt. The PR curve involves more difficult statistical concepts and could have been difficult to read for people not familiar with these methods, making the article less comprehensible. The other methods used (data splitting and resampling) ensure the performance obtained in the train database.

  1. Figure 2. What does se, spe, mean? Please define it in figure legends.

The definitions of Se and Spe have been included in Figure legends. Se: sensitivity; Spe: specificity

  1. Please describe in the main document the 3 Influence of sex on the PPi cutoff for PXE diagnosis.

The data corresponding to the influence of sex on the PPi cutoff for PXE diagnosis have been given in the last part of the results section and in the Materials and Methods section. The cutoff value was not influenced by sex: this information was also given in abstract and discussion sections.

  1. The discussion section needs to be improved.

The discussion has been improved according to the referee’s suggestion.

  1. Limitations: in addition to the limitations, it must be included that the number of patients is small.

The limitation of this study by the small size of the cohort (PXE is a rare disease) is discussed at the end of the discussion.

  1. The conclusion should be improved. The authors mention “significantly contribute to the development and validation of future treatments for this disease.” Please explain how these results contribute to future treatments for this disease.

We have improved the conclusion according to referee comments and we have cited one clinical trial currently in development by our team.

  1. The authors should mention in the introduction and results that they used combinations of enzymatic and ion chromatography techniques as a cross-control.

We paid attention to this comment and mentioned the information that we used combinations of enzymatic and ion chromatography techniques as a cross-control in the introduction and results.

Reviewer 2 Report

Comments and Suggestions for Authors

The authors aimed to figure out an accurate cutoff value of PPi in PXE patient in order to develop a novel diagnosis method for this rare genetic disease that contribute in development of future treatment. The manuscript is clear and well structured. the cited references are recent. The experimental design is appropriate with good sample size for this rare disease. the figures are understandable and include clear details and stats. The conclusion is consistent with the evidence and arguments presented. 

I have only one point that authors need to clarify  , according to Van Gils, Matthias et al., 2023 ( authors cited this study) PPi level doesn't correlate with the genotype or phenotype of this disease. The authors should mention that in the discussion section and clarify why they used this parameter  as a dignosis method inspire of this cited study. 

Author Response

First, we would like to thank the two reviewers who took time to assess our manuscript and for the critical and constructive feedback they provided. Following those comments, we revised our manuscript to address their concerns and we hope that this study is now suitable for publication in IJMS. Changes are marked up in red in the revised version and a point-by-point response to the reviewers is provided below in blue.

Reviewer 2

Comments and Suggestions for Authors

The authors aimed to figure out an accurate cutoff value of PPi in PXE patient in order to develop a novel diagnosis method for this rare genetic disease that contribute in development of future treatment.

The manuscript is clear and well structured. the cited references are recent. The experimental design is appropriate with good sample size for this rare disease. the figures are understandable and include clear details and stats. The conclusion is consistent with the evidence and arguments presented. 

We thank the reviewer for his positive comments.

I have only one point that authors need to clarify, according to Van Gils, Matthias et al., 2023 (authors cited this study) PPi level doesn't correlate with the genotype or phenotype of this disease. The authors should mention that in the discussion section and clarify why they used this parameter as a diagnosis method inspire of this cited study. 

A recent study pointed out that PPi level doesn't correlate with the genotype or phenotype of PXE disease (Van gils et al. 2023). However, we have reported a strong correlation between age, arterial calcification, and disease severity (Leftheriotis et al., 2022) suggesting that time of exposure to a low plasma PPi concentration is a major determinant of arterial calcification and clinical severity in PXE. However, a low PPi is likely to explain a large part of the calcifying phenotype and thus could represent a diagnostic criterion as suggested by the excellent discriminative power of the plasma PPi assayed by our method.

Round 2

Reviewer 1 Report

Comments and Suggestions for Authors

I thank the authors for considering the comments. The manuscript is substantially improved and can be accepted.